# Antibacterial, Antifungal and Anticancer Activities of Compounds Produced by Newly Isolated *Streptomyces* Strains from the Szczelina Chochołowska Cave (Tatra Mountains, Poland)

**DOI:** 10.3390/antibiotics10101212

**Published:** 2021-10-05

**Authors:** Weronika Jaroszewicz, Patrycja Bielańska, Daria Lubomska, Katarzyna Kosznik-Kwaśnicka, Piotr Golec, Łukasz Grabowski, Ewa Wieczerzak, Weronika Dróżdż, Lidia Gaffke, Karolina Pierzynowska, Grzegorz Węgrzyn, Alicja Węgrzyn

**Affiliations:** 1Department of Molecular Biology, Faculty of Biology, University of Gdansk, Wita Stwosza 59, 80-308 Gdansk, Poland; weronika.jaroszewicz@ug.edu.pl (W.J.); p.bielanska.606@studms.ug.edu.pl (P.B.); darialajn@gmail.com (D.L.); weronika.drozdz@student.uj.edu.pl (W.D.); lidia.gaffke@ug.edu.pl (L.G.); karolina.pierzynowska@ug.edu.pl (K.P.); grzegorz.wegrzyn@biol.ug.edu.pl (G.W.); 2Laboratory of Phage Therapy, Institute of Biochemistry and Biophysics, Polish Academy of Sciences, Kładki 24, 80-822 Gdansk, Poland; k.kwasnicka@hotmail.com (K.K.-K.); lukas.grabowski95@gmail.com (Ł.G.); 3Department of Molecular Virology, Institute of Microbiology, Faculty of Biology, University of Warsaw, Miecznikowa 1, 02-096 Warsaw, Poland; pgolec@biol.uw.edu.pl; 4Department of Biomedical Chemistry, Faculty of Chemistry, University of Gdansk, Wita Stwosza 63, 80-308 Gdansk, Poland; ewa.wieczerzak@ug.edu.pl; 5Faculty of Biochemistry, Biophysics and Biotechnology, Jagiellonian University, Gronostajowa 7, 30-387 Krakow, Poland

**Keywords:** cave microorganisms, *Actinobacteria*, antibacterial activity, antifungal activity, anticancer activity

## Abstract

Resistance of bacteria, fungi and cancer cells to antibiotics and other drugs is recognized as one of the major problems in current medicine. Therefore, a search for new biologically active compounds able to either kill pathogenic cells or inhibit their growth is mandatory. Hard-to-reach habitats appear to be unexplored sources of microorganisms producing previously unknown antibiotics and other molecules revealing potentially therapeutic properties. Caves belong to such habitats, and *Actinobacteria* are a predominant group of microorganisms occurring there. This group of bacteria are known for production of many antibiotics and other bioactive compounds. Interestingly, it was demonstrated previously that infection with bacteriophages might enhance production of antibiotics by them. Here, we describe a series of newly isolated strains of *Actinobacteria* that were found in caves from the Tatra Mountains (Poland). Phage induction tests indicated that some of them may bear active prophages able to produce virions upon treatment with mitomycin C or UV irradiation. Among all the examined bacteria, two newly isolated *Streptomyces* sp. strains were further characterized to demonstrate their ability to inhibit the growth of pathogenic bacteria (strains of *Staphylococcus aureus, Salmonella enterica, Enterococcus* sp., *Escherichia coli,* and *Pseudomonas aeruginosa*) and fungi (different species and strains from the genus *Candida*). Moreover, extracts from these *Streptomyces* strains reduced viability of the breast-cancer cell line T47D. Chemical analyses of these extracts indicated the presence of isomers of dichloranthrabenzoxocinone and 4,10- or 10,12-dichloro-3-O-methylanthrabenzoxocinone, which are putative antimicrobial compounds. Moreover, various previously unknown (unclassified) molecules were also detected using liquid chromatography–mass spectrometry, suggesting that tested *Streptomyces* strains may synthesize a battery of bioactive compounds with antibacterial, antifungal, and anticancer activities. These results indicate that further studies on the newly isolated *Actinobacteria* might be a promising approach to develop novel antibacterial, antifungal, and/or anticancer drugs.

## 1. Introduction

Resistance of pathogenic bacteria to most, if not all, antibiotics available for medical use is one of the major problems in current medicine. This apparent antimicrobial agent-resistance crisis may lead to serious health problems, with millions of fatal cases of bacterial infections, if no effective actions are conducted to solve this problem [1]. Selection of antibiotic-resistant bacterial strains has been ascribed mainly to the misuse of antibiotics in medicine and animal farming [2]. However, irrespective of the actual cause of appearance of many multiple antimicrobial-resistant bacterial strains, there is an urgent need to find novel ways to treat bacterial infections effectively [3]. It is estimated that total cost of antibiotic resistance is currently as high as USD 55 billion per year worldwide, and this cost may increase up to USD 100 trillion by 2050; this can be accompanied by about 10 million death cases per year caused by infections with antibiotic-resistant microbes [1].

In this light, it is crucial to take intensive actions to prevent the putative scenario described above. The World Health Organization has presented an action plan to solve this problem, which is based on understanding of mechanisms of antibiotic resistance, strengthening the knowledge through extensive research, reducing incidence of infection, optimizing the use of antibiotics, and ensuring investment in countering antimicrobial resistance [1]. Several approaches can be proposed to find novel ways to combat infections caused by pathogenic bacteria. Among them, intensification of vaccination, the use of bacteriophages or products of expression of their genes, the use of herbal products, and searching for novel antibiotics appear to be the most promising options [3]. 

One may assume that discovery of novel antimicrobial molecules is possible mainly by exploring rarely investigated environments [4]. These include marine waters, glaciers, hot springs, underground lakes, hydrothermal vents, and caves. In fact, such a strategy might be effective; however, technical difficulties related to investigation of hardly accessible habitats may be a serious drawback for attempts to isolate previously unknown microorganisms and compounds produced by them [4]. On the other hand, this can be one of a very few options to solve the antimicrobial-resistance crisis. Therefore, in this work we focused on isolation of microbial strains from a hardly accessible mountain cave and tested if such strains can produce previously unknown compounds revealing strong antimicrobial and/or anticancer activities.

Previous attempts to find antimicrobial compounds produced by microorganisms occurring in caves were relatively rare but indicated a strong potential, despite difficulties in obtaining the biological material and in culturing newly isolated strains. It was found that actinomycetes are predominant microorganisms isolated from caves [5]. When exploring caves, bacteria could be isolated from rock wall [6], cave soil [7], sediment [8], water [8], and moonmilk deposits [9]. Isolation and characterization of microbial strains from caves, including their antibacterial, antifungal, anti-inflammatory, antioxidative, and anticancer activities, and identification of some specific compounds responsible for such biological actions were reviewed recently [4]. These strains and compounds are promising. However, difficulties in cultivation of these microorganisms as well as specificity of detected compounds to some bacterial or fungal species indicate that further studies on a higher number of isolates are reasonable. Interestingly, it was reported that antibiotic production by *Streptomyces venezuelae* strain ISP5230 can be enhanced under various environmental conditions, such as elevated temperature and ethanol treatment, as well as after infection with bacteriophage SV1 [10]. Moreover, characterization of bacteriophage phiSASD1-endoced endolysin and holin, as potential antibacterial drugs, was reported recently [11]. Therefore, one might suggest that this phage, infecting *Streptomyces avermitilis*, can be an important source of genes encoding newly identified antibacterial agents, active against various pathogenic strains of *Staphylococcus aureus*, *Sarcina lutea*, and *Enterococcus faecalis* [11]. The studies summarized above encouraged us to search for previously unknown microbes, derived from caves, that might reveal various antibacterial, antifungal, and anticancer properties.

## 2. Results

### 2.1. Isolation and Identification of Bacteria from the Szczelina Chochołowska Cave

Using samples of water and moonmilk from the Szczelina Chochołowska cave (Tatra Mountains, Poland; Figure 1), we obtained 24 isolates of bacteria. These isolates were cultured under laboratory conditions, and on the basis of DNA isolation and sequencing of the 16S rRNA gene, they were identified at the genus or species level. All of them belong to *Actinobacteria*, and to the following genera: *Arthrobacter*, *Frigoribacterium*, *Microbacterium, Nocardia*, *Nocardiopsis*, *Rhodococcus*, *Streptomyces*, and *Tomitella* (Table 1).

### 2.2. Prophage Induction and Determination of the Presence of Phage Virions

Since bacteriophage presence has been reported previously as a factor enhancing production of antibiotics by *Actinobacteria* [10], we tested whether the isolated strains contain inducible prophages. Cultures of the isolates were either treated with 0.5 μg/mL mitomycin C or UV-irradiated, and following further cultivation and centrifugation, supernatants were tested for the presence of phage virions using electron microscopy. Bacteriophage virions were found in samples derived from the *Nocardia* sp. strain JSZCL7 (Figure 2). However, we were not able to propagate the isolated phage under laboratory conditions. The host strain of Nocardia could not be effectively infected with this phage, most probably due to immunity of the lysogen, and we could not find a non-lysogenic strain sensitive to this phage. Moreover, virions appeared very fragile under laboratory conditions and in buffers employed in this study, which could be observed as damaged or disconnected heads and tails (Figure 2). Thus, we failed to obtain single plaques of this phage and to propagate it for further analyses. Nevertheless, the presence of bacteriophages in samples of isolated bacteria may suggest that this group of viruses might be taken into consideration in further studies on bioactive compound-producing *Actinobacteria*, as putative modulators of syntheses of such substances.

### 2.3. Preliminary Screening for Production of Antibacterial Compounds by the Isolates and Selection of Strains for Further Investigation

To test abilities of the actinobacterial isolates from the Szczelina Chochołowska cave to produce antimicrobial compounds, we used a battery of pathogenic bacterial and fungal strains (Table 2). The preliminary screening, using the streak-test, indicated that 3 isolates revealed the highest antibacterial and antifungal properties when contacting other microorganisms (estimated as the number of strains in which growth was inhibited). These isolates were M2_9, M4_24, and M5_8, all belonging to *Streptomyces* sp. (Table 1). Analyses of 16S rDNA sequences revealed that this fragment of the genome is identical in M2_9 and M5_8, thus only one of these strains (M5_8) was investigated further for antimicrobial activities. We also performed phylogenetic analyses, based on 16S rDNA, which indicated that M4_24 and M5_8 are closely related strains (Figure 3). 

### 2.4. Antibacterial and Antifungal Activities of Streptomyces M4_24 and M5_8 Strains

To assess antibacterial and antifungal activities of *Streptomyces* M4_24 and M5_8 strains, we performed a streak-test and measured the width of growth inhibition zones of various pathogenic strains of bacteria and fungi after contacting the newly isolated *Actinobacteria*, as indicated in Appendix A. 

We found that M4_24 and M5_8 isolates revealed antimicrobial activities against most tested strains of bacteria (Table 3) and fungi (Table 4), while M4_24 was generally more effective than M5_8 when considering the number of strains in which growth was halted.

### 2.5. Anticancer Activities of Streptomyces M2_9, M4_24 and M5_8 Isolates 

To determine if the *Streptomyces* isolates produce anticancer compounds, we tested the effects of extracts of cell cultures on the viability of breast-cancer cells (the T47D cell line). To obtain the extracts, *Actinobacteria* were cultured in liquid media with various pH values. We found that viability of the tested cancer cells decreased significantly with increasing concentrations of the extracts from investigated *Actinobacteria*, irrespective of the pH value of the medium used for *Streptomyces* cultivation (Figure 4). In control experiments, we did not observe any significant reduction of viability of non-transformed cells, the HDFa cell line, except conditions of 100% extract (results not shown). Therefore, to test if the observed effects on the breast cancer cells are caused by the compounds present in the extract rather than from dilution of eukaryotic cell-culture medium by the extract, we performed control experiments in which the medium used for cultivation of *Actinobacteria* was used instead of the extract. When we compared effects of the *Actinobacteria*-free medium with the analogous medium but derived from Streptomyces cultures, we found that inhibition of viability of T47D cells was specific for the extracts of actinobacterial cultures (Figure 5). 

### 2.6. Chemical Analyses of Extracts from Cultures of Streptomyces M2_9, M4_24 and M5_8 Isolates 

We aimed to test which compounds, potentially responsible for antibacterial, antifungal and anticancer activities of *Streptomyces* M2_9, M4_24 and M5_8 isolates, are produced by the investigated *Actinobacteria*. Therefore, we analyzed extracts of cultures of these bacteria. All analyzed crude extracts presented similar chemical profiles (Figure 6). The screening for known bioactive metabolites in the extracts, using the *Dictionary of Natural Products*, revealed that the analyzed isolates can produce many natural products previously not detected in *Streptomyces* spp. Only two compounds could be preliminarily predicted, using search parameters described in Section 4.10. These compounds were identified as the isomers of dichloranthrabenzoxocinone (4,10-; 4,12- or 10,12-; *m*/*z* 530.344; Figure 7) and 4,10- or 10,12-dichloro-3-O-methylanthrabenzoxocinone (*m*/*z* 544.360; Figure 8). Their formulas are presented in Figure 9. They are described to exhibit moderate antibacterial activities [12]. Other metabolites found in the analyzed extracts appear to be novel, which supports the potential of cave *Actinobacteria* as the producers of previously unknown bioactive substances.

## 3. Discussion

Since microorganisms occurring in hard-to-reach environments were indicated previously as a potent, though still unexplored, source of bioactive compounds, including antibacterial, antifungal, and anticancer substances [4], in this work we aimed to isolate bacteria living in one of the caves that were not investigated to date for this purpose, and to test their potential in producing compounds with useful properties. We chose the Szczelina Chochołowska cave, located in the Tatra Mountains (Poland), and characterized 24 microbial isolates that were subsequently tested for their antibacterial, antifungal, and anticancer activities. All the isolates belong to *Actinobacteria*, which is not a surprise as it was demonstrated previously that this group of microorganisms is predominant in caves [5]. Since previous studies indicated that bacteriophages may significantly modulate production of antimicrobial agents by *Streptomyces* [10], we tested if the isolates are lysogenic for bacteriophages. In fact, we were able to induce a prophage from *Nocardia* sp. strain JSZCL7 and to demonstrate that phage virions can be formed. However, we were not able to find a host in which this bacteriophage could be propagated. The lysogenic strain could not be infected with the same phage, and it is likely that the isolated bacteriophage may be of narrow host range. Moreover, this bacteriophage appeared very fragile under laboratory conditions, which further caused problems with its characterization. Therefore, although we demonstrated the presence of inducible prophages in *Actinobacteria* isolates from the Szczelina Chochołowska cave, we could not determine whether these viruses are able to modulate production of bioactive compounds by their host cells. 

We found that the *Streptomyces* isolates revealed antibacterial and antifungal activities against various pathogenic strains, as indicated by the streak-test. The inhibition of growth of most of these strains was evident after a contact with either the M4_24 or M5_8 isolate, though the former one was effective against more bacterial and fungal strains. Extracts from cultures of these isolates could also reduce viability of breast-cancer cells (T47D line). Importantly, chemical analyses of such extracts indicated the presence of 4,10-dichloroanthrabenzoxocinone, 10,12-dichloroanthrabenzoxocinone, 4,12-dichloroanthrabenzoxocinone, 4,10-dichloro-3-O-methylanthrabenzoxocinone, and 10,12-dichloro-3-O-methylanthrabenzoxocinone, compounds that are potential antibiotics. In addition, various unknown compounds were also detected, suggesting that a set of novel bioactive molecules produced by *Streptomyces* M4_24, M2_9, and M5_8 isolates is even larger. 

Our results confirmed that *Actinobacteria* isolated from caves may be a rich source of potential antibiotics. Examples of previous works in this field include isolation of cervamicin A-D from *Streptomyces tendae* HKI 0179 [13], isolation of strains inhibiting the growth of various Gram-positive [14,15] and Gram-negative [16] bacteria, and identification of undecylprodigiosin, produced by *Streptomyces* sp. JS520 [17]. In fact, several reports demonstrated that most of the bacterial strains isolated from caves are able to inhibit growth of other bacteria and/or fungi, though only a few active compounds have been identified, such as pyrrolopyrazines pyrrolo[1,2-a]pyrazine-1,4-dione, hexahydro-3-(2-methylpropyl), pyrrolo[1,2-a]pyrazine-1,4-dione, hexahydro-3-(phenylmethyl), and 1,2-benzenedicarboxylic acid, bis(2-methylpropyl) ester [6,7,8,9,18,19,20,21,22,23,24]. Moreover, anticancer activities of compounds derived from strains of bacteria occurring in caves were also reported previously. These compounds include hypogeamicin A, xiakemycin A, huanglongmycin A, and various unidentified molecules [25,26,27].

It seems that anthrabenzoxocinone compounds might be of special interest among substances isolated from *Streptomyces* strains. Early studies on *Streptomyces violaceusniger*, isolated in Japan, led to discovery of a potent anthrabenzoxocinone derivative, named BE-24566B, which might be used against MRSA [28]. Two such compounds were then isolated from *Streptomyces* sp. (MA6657), and their characteristics indicated that they have significant antimicrobial activities against Gram-negative bacteria, with MIC values ranging from 0.5 to 2.0 μg/mL [12]. Genetic analysis of *Streptomyces* sp. FJS31-2 indicated that its genome contains a gene cluster that is responsible for production of anthrabenzoxocinones with potential antibacterial activities, previously known BE-24566B and newly identified zunyimycin A [29]. The same *Streptomyces* strain, when cultured under various laboratory conditions, produced other compounds from this group, named zunyimycins B and C, inhibiting growth of both MRSA and enterococci [30]. These results demonstrated a high potential of *Actinobacteria* to produce different antimicrobial compounds that may be modified under different environmental conditions. In addition, antibiotic activities of natural anthrabenzoxocinones can be enhanced by biochemical modifications [31]. Furthermore, recent studies highlighted a high biodiversity of natural products, including anthrabenzoxocinone compounds, produced by bacteria isolated from natural habitats [32], and a high genetic potential of *Streptomyces* spp. in production of a variety of bioactive compounds, also including anthrabenzoxocinones [33]. Our results, presented in this report, corroborate these conclusions.

In summary, cave-derived *Actinobacteria* reveal various antibacterial, antifungal, and anticancer activities, while among them only relatively few biologically active compounds were identified. This indicates that caves are habitats rich in microorganisms producing as-yet unknown substances that might be potentially used in treatment for various infections and/or cancers. Definitely, further more detailed studies are required in this field, and our work fits to this topic, providing further evidence for effectiveness of *Actinobacteria* isolated from caves in inhibiting growth of pathogenic bacteria and fungi, and reducing viability of cancer cells. Whether modulation of production of bioactive compounds by bacteriophages (as reported previously [10]) is a specific or a more general phenomenon remains to be elucidated.

## 4. Materials and Methods

### 4.1. Bacterial Strains

Newly discovered isolates of bacteria from the Szczelina Chochołowska cave are presented in Table 1. Pathogenic bacteria and fungi used in this work were from various sources that are presented in Table 2, together with characteristics of these strains. All *Salmonella enterica* serotypes were from National Salmonella Center in Gdańsk (Poland). *Staphylococcus aureus* strains were from the Department of Medical Microbiology, Medical University of Gdansk (Poland) [34]. *Pseudomonas aeruginosa*, *Bacillus* spp. and Shiga-toxin producing *Escherichia coli* were from the collection of the Department of Molecular Biology, University of Gdańsk (Poland). Fungal strains were from Bruss Laboratories, Gdynia (Poland) and University Medical Center of Medical University of Gdańsk, Gdańsk (Poland).

### 4.2. Cave Description and Sampling

The Szczelina Chochołowska cave is located in the Tatra National Park (TNP; Poland) in Western Tatra. It is situated orographic left slope of the Valley Chochołowska (19°48’43″.140 E 49°14’45″.401 N) (WGS84 coordinates). Szczelina Chochołowska has 2320 m of cave passages and three entrances (1-E exposition at 1051 m a.s.l; 2-SE at 1072 m a.s.l; 3-NE at 1083 m a.s.l), with 60 m of height difference (Figure 1).

Samples of water and moonmilk were collected from the six parts of cave (Figure 1B–D), according to the permission of the Minister of the Environment (Poland) (DLP-III.286.102.2016.MGr) and TNP director (DBN.505/14/15 RÓŻ no 128, DBN.505/14/16 RÓŻ no 128, DBN.505/14/17 RÓŻ no 128). Sampling sites mainly depended on water flow and the presence of speleothems, such as moonmilk deposits. The first sampling site was located relatively close to the entrance (Figure 1A); samples were taken from the ice formations. Five samples were taken from the selected sites, such as small ponds (Figure 1B, sampling sites 3, 4, 5, 8; Figure 1D, sampling site 13), moonmilk (Figure 1B, sampling sites 4, 7, 10; Figure 1C, sampling site 11; Figure 1D, sampling site 12), and water dripping from speleothems (Figure 1C, sampling sites 2, 7, 6, 9). All samples were collected using sterile tubes and disposable pipettes. Withdrawn samples were transported to the Department of Molecular Biology of University of Gdansk, Gdańsk (Poland) laboratories in a cooler packed with ice, and kept in 4 °C until cultivating.

### 4.3. Bacterial Growth Conditions

Samples of water and moonmilk (Section 4.2) were refrigerated at 4 °C. Serial dilutions (10, 100, and 1000 times in sterile, distilled water) were prepared and plated using Reasoner’s 2A (R2A) agar medium [35] supplemented with water collected from the Szczelina Chochołowska cave. The plates were incubated at either room temperature or 4 °C for 14 days. Isolated colonies were maintained without access to the light on the R2A agar plates at 4 °C for subsequent studies, and glycerol stocks (30% *v*/*v*) were kept in a deep-freezer (at −80 °C) for long-term preservation. Liquid cultures were prepared in the R2A liquid medium [35].

Bacterial strains listed in Table 2, when used alone, were cultured in LB liquid or agar media [36].

### 4.4. Prophage Induction and Electron Microscopy of Bacteriophage Virions

Liquid cultures in the R2A medium, supplemented with 10 mM MgSO_4_ and 10 mM CaCl_2_, were incubated to OD_600_ ~0.3. Mitomycin C was added to 0.5 μg/mL and the incubation was prolonged for 24 h. Alternatively, 25 mL of the culture was transferred to a Petri dish, and irradiated with UV (at 320 nm wavelength) for 5 s, and incubated as described above. Then, the culture was centrifuged (8000× *g*, 10 min, room temperature), and the supernatant was treated with DNase I (2 μg/mL) and RNase A (2 μg/mL) for 30 min. Polyethylene glycol was added to final concentration 10%, and the mixture was stirred for 24 h at 4 °C. Following centrifugation (10,000× *g*, 10 min, 4 °C), the pellet was suspended in TM buffer (10 mM Tris-HCl, 10 mM MgSO_4_, pH 7.2), and filtered through 0.22 μm microbiological filter. After triple extraction with 0.33 volume of chloroform and centrifugation (8000× *g*, 5 min, room temperature), the lysate was kept at 4 °C.

Transmission electron microscopic analyses of phage visions were conducted as described previously [37]. Briefly, negatively stained (with uranyl acetate) virions were observed and photographed under a Philips CM 100 electron microscope.

### 4.5. Streak-Test

Tested isolates of *Actinobacteria* were streaked perpendicularly on R2A agar plates and left to grow in the dark at room temperature for 48 or 72 h. Pathogenic bacteria and fungi were then streaked diagonally onto the plates with grown isolates and left for 24 h in the dark at room temperature. Growth-inhibition zones were measured after incubation. Each experiment was performed in triplicate.

### 4.6. Molecular Identification of Isolates and Phylogenetic Analyses

Identification of isolates was based on the molecular analysis of 16S rRNA gene sequences. Whole genome DNA was extracted using an Ultraclean Microbial DNA Isolation Kit (MO BIO, Carlsbad, CA USA) following the manufacturer’s protocol. DNA concentration was determined using Nanodrop ND-1000 Spectrophotometer and agarose gel electrophoresis. Sequences of 16S rRNA genes were PCR-amplified with oligonucleotides 785F/907R and sequenced by Macrogen Inc. (Amsterdam, The Netherlands). The sequences were checked for potential chimeric artifacts using the DECIPHER online tool. The sequences obtained in this study were deposited in GenBank with accession numbers KU643201.1, KU643207.1, MG758033.1, and MG758033.1. Sequences were compared to NCBI GenBank database using BLASTn to identify the closest relatives based on 16S rRNA sequences. The alignment and a Neighbor-Joining (NJ) tree [38], based on the Jukes–Cantor Genetic Distance Model [39], was constructed using the MEGA X software [40], with *Saccharopolyspora erythrea* NRRL 2338 as an outgroup. Graphical processing was conducted with Inkscape 0.92.4.

### 4.7. Preparation of Extracts from Cultures of Isolates

Material from colonies of *Streptomyces* isolates were streaked onto multiple R2A agar plates and left to grow in the dark at room temperature for 5 days. The R2A agar with grown bacteria was then cut into stripes and suspended in 2 L of the liquid R2A medium. The mixture was shaken and incubated in the dark, at room temperature, for 14 days. The mixture was stirred every 2 days in order to enable bacteria and metabolites to diffuse into liquid medium.

### 4.8. Cancer and Non-Transformed Cell Lines and Cell Cultures

Breast-cancer cell line T47D [41] was purchased from Sigma Aldrich (Darmstadt, Germany), and used for cell-culture experiments. HDFa cell line [42] was used as a control of non-transformed cells. The cells were cultured at 37 °C in a humidified atmosphere with 5% CO_2_, in the DMEM medium, containing the penicillin–streptomycin mixture, and supplemented with 10% fetal bovine serum, as described previously [42].

### 4.9. Estimation of Cells’ Viability

Viability of eukaryotic cells was estimated as described previously [43], using the MTT (3-(4,5-dimethylthiazol-2-yl)-2,5-diphenyltetrazolium bromide) test. Briefly, 3 × 10^3^ cells were passaged wells of the 96-well plate and incubated overnight. Extracts of *Streptomyces* cultures were added to indicated final concentrations, and the incubation prolonged for another 24 h at 37 °C. Following addition of 25 µL of the 4 mg/mL MTT solution to each well, and 3-h incubation at 37 °C, 100 µL of DMSO was added to dissolve formazan crystals. Metabolic activity of cells was estimated by measurement of absorbance at 570 nm and 620 nm (using Victor3 microplate reader) and comparison to control samples (untreated cells, i.e., 0% extract). Each experiment was repeated 3 times. Statistical significance of differences between results of experiments with extracts and controls was tested using ANOVA with a Tukey post hoc test. The differences were considered significant when *p* < 0.05.

### 4.10. Chemical Analyses

In order to extract secondary metabolites, each liquid culture of *Streptomyces* isolate was poured into a flask with an equal volume of ethyl acetate and agitated overnight at 100 rpm at room temperature. The organic phase was separated and dried over anhydrous MgSO_4_, and the solvent was evaporated on a rotatory evaporator.

Crude extracts were analyzed by liquid chromatography UHPLC (Nexera-i, Shimadzu) with a Kinetex-C8 column (2.1 mm × 100 mm, 2.6 µm, 100 Å) using 15 min linear gradient of 5–100% B (B-80% acetonitrile) in 0.1% aqueous trifluoroacetic acid, and liquid chromatography–mass spectrometry, using LC-MS-IT-TOF (Shimadzu, Kyoto, Japan) with a Kromasil-C8 column (1 mm × 250 mm, 5 µm, 90 Å). The mass detection was performed in positive ion mode.

The screening for known compounds was performed using the *Dictionary of Natural Products* database, version 30.1 (https://dnp.chemnetbase.com/faces/chemical/ChemicalSearch.xhtmlm, accessed on 14 September 2021) with the following search parameters: biological source of natural product and the accurate molecular mass. Compounds were considered to be preliminarily identified when the difference in accurate mass was lower than 0.05.

## Figures and Tables

**Figure 1 antibiotics-10-01212-f001:**
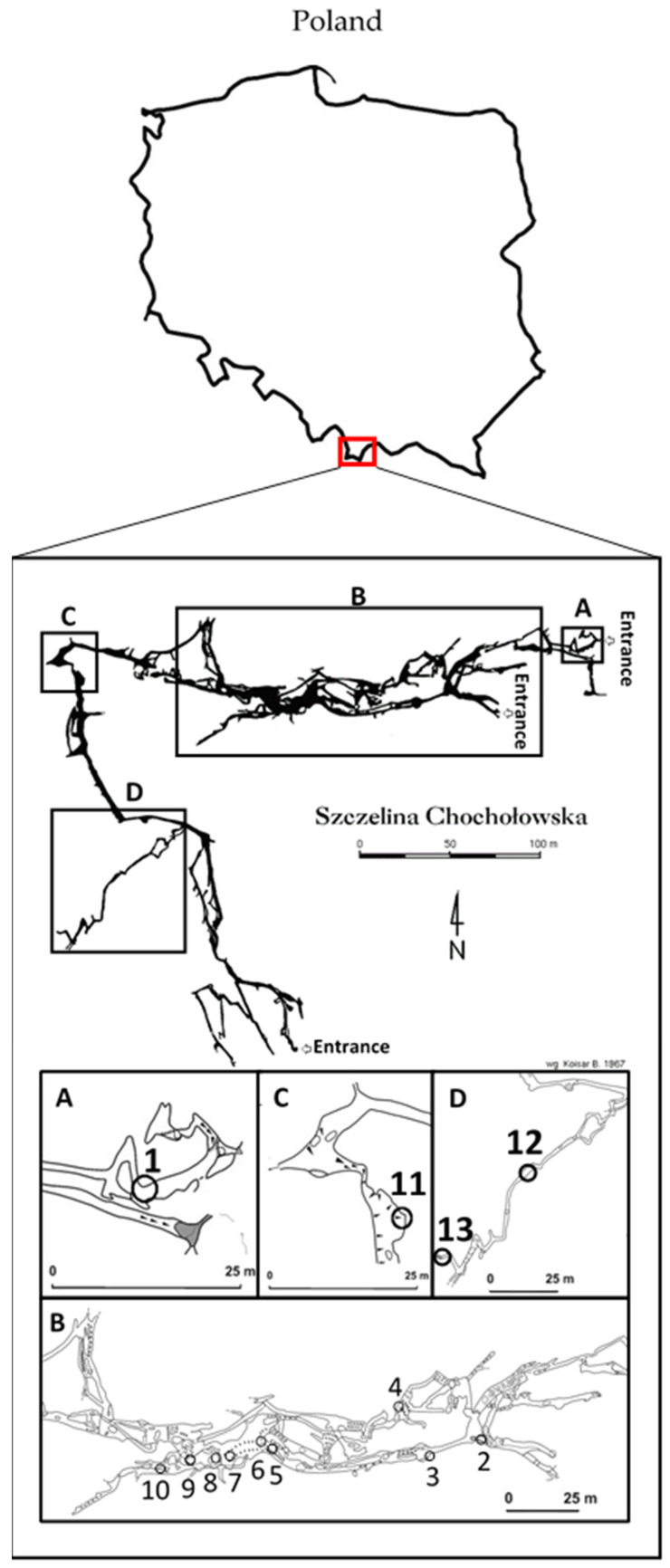
A map of the Szczelina Chochołowska cave (19°48’43″.140 E 49°14’45″.401 N) (WGS84 coordinates). A map of Poland is shown at the top of the figure with the square indicating the region in which the cave is located. Regions marked as (**A**–**D**) are enlarged at the bottom of the figure. Numbers indicate places of collection of samples. Details are described in Section 4.2.

**Figure 2 antibiotics-10-01212-f002:**
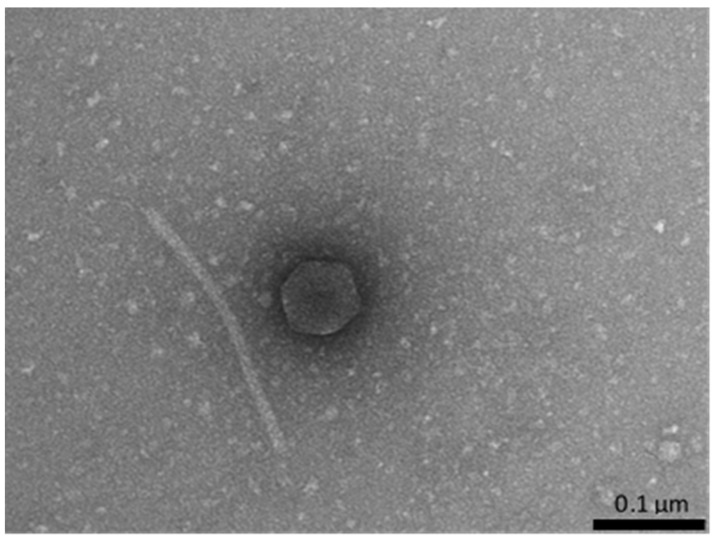
Electron micrograph of the virion of a bacteriophage isolated after induction of the *Nocardia* sp. strain JSZCL7 with 0.5 μg/mL mitomycin C. Virions were very unstable under laboratory conditions, which is exemplified by disconnected head and tail of the virion.

**Figure 3 antibiotics-10-01212-f003:**
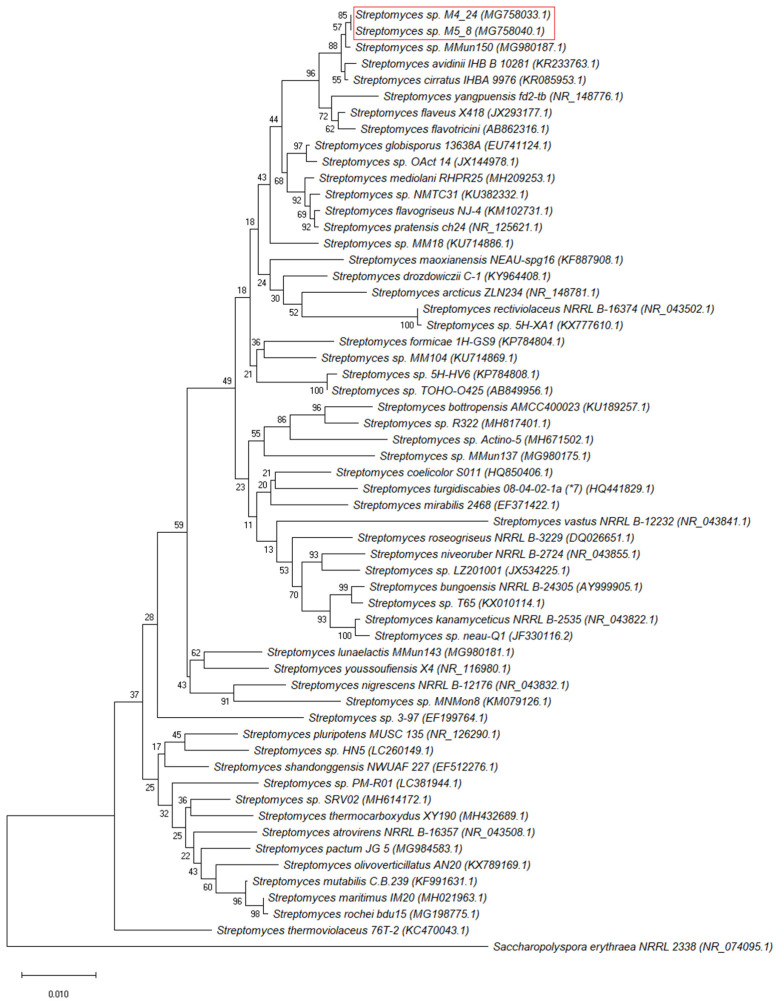
Phylogenetic analysis of *Streptomyces* M4_24 and M5_8 strains. The alignment and a Neighbor-Joining (NJ) tree, based on Jukes–Cantor Genetic Distance Model, was constructed using the MEGA X software, with *Saccharopolyspora erythrea* NRRL 2338 as an outgroup. Bootstrap values are shown from 1000 replicates.

**Figure 4 antibiotics-10-01212-f004:**
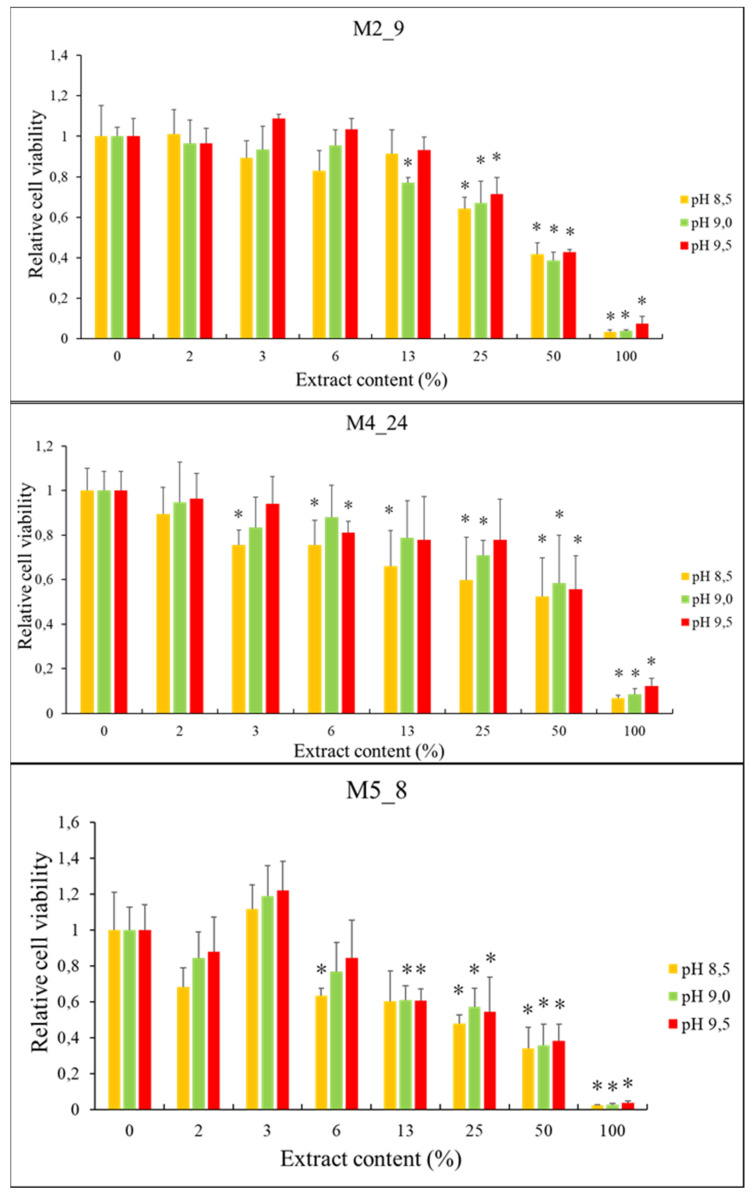
Effects of extracts from cultures of *Streptomyces* M2_9, M4_24, and M5_8 isolates on viability of T47D cells (assessed by the MTT test). Mean values from 3 independent experiments ± SD are demonstrated. Asterisks (*) indicate statistically significant differences (*p* < 0.05) relative to results obtained for samples with no extract (0% extract content).

**Figure 5 antibiotics-10-01212-f005:**
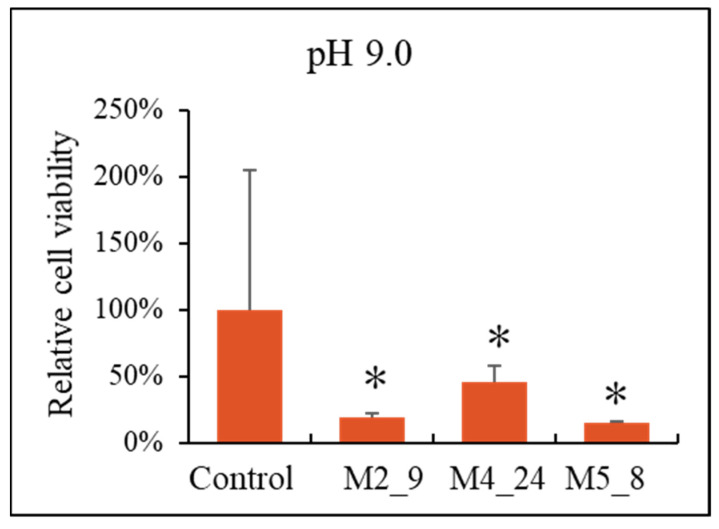
Effects of the bacterial culture medium alone (control, the value assumed to be 100%) and extracts from cultures of *Streptomyces* M4_9, M4_24, and M5_8 isolates on viability of T47D cells (assessed by the MTT test). Mean values from 3 independent experiments ± SD are demonstrated. Asterisks (*) indicate statistically significant differences (*p* < 0.05) relative to the control.

**Figure 6 antibiotics-10-01212-f006:**
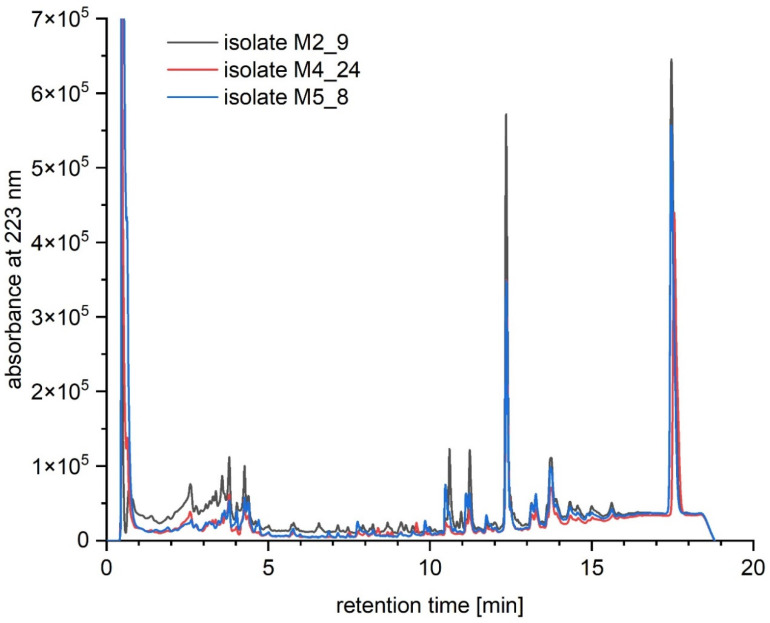
Overlayed chromatograms obtained by UHPLC of crude extracts of cultures of *Streptomyces* isolates M2_9 (black), M4_24 (red), M5_8 (blue).

**Figure 7 antibiotics-10-01212-f007:**
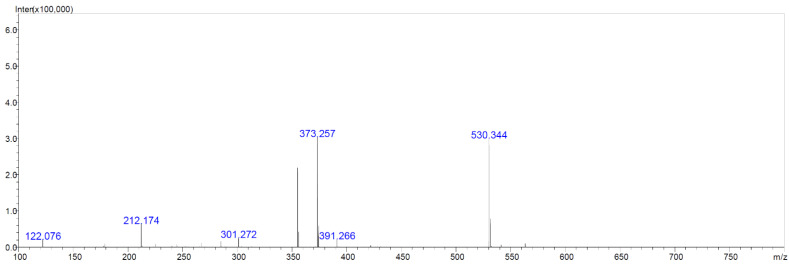
LC-MS chromatogram of the isomer of dichloroanthrabenzoxocinone (M = 529.366 Da).

**Figure 8 antibiotics-10-01212-f008:**
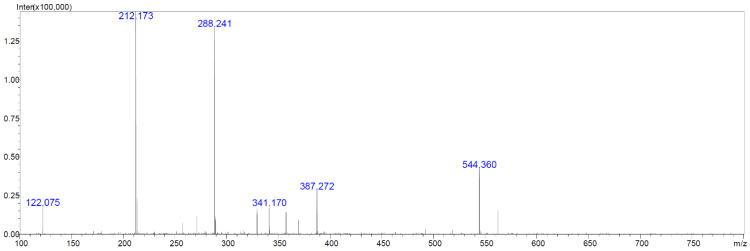
LC-MS chromatogram of the isomer of dichloro-3-O-methylanthrabenzoxocinone (M = 543.393 Da).

**Figure 9 antibiotics-10-01212-f009:**
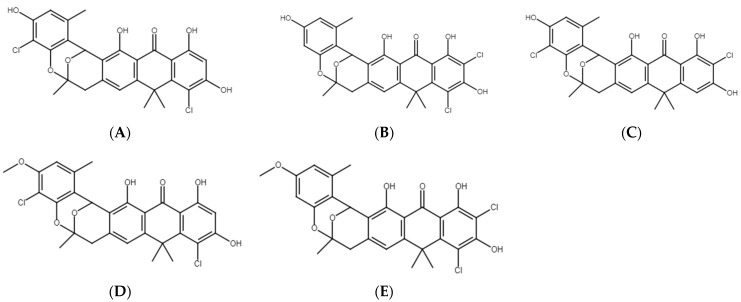
Chemical structures of the preliminarily detected natural products produced by *Streptomyces* isolates M2_9, M4_24, and M5_8. (**A**) 4,10-Dichloroanthrabenzoxocinone, (**B**) 10,12-Dichloroanthrabenzoxocinone, (**C**) 4,12-Dichloroanthrabenzoxocinone, (**D**) 4,10-Dichloro-3-O-methylanthrabenzoxocinone, (**E**) 10,12-Dichloro-3-O-methylanthrabenzoxocinone.

**Table 1 antibiotics-10-01212-t001:** *Actinobacteria* isolates from the Szczelina Chochołowska cave, with identification of the closest relatives determined on the basis of comparison of 16S rDNA sequences.

Isolate ^1^	Strongest 16S rDNA Sequence Match (BLASTN)			
	Organism	Accession No. ^2^	Bits	%
JHARAB1_N	*Arthrobacter* sp. strain VTT E-052904	EF093123	2691	99.9
JHARN2	*Rhodococcus* sp. strain UFZ-B528	AF235012	2667	99.9
JSZCO2	*Microbacterium* sp. strain JSZCO2	KU643207	2705	100
JSZCZL7	*Nocardia* sp. strain JSZCL7	KU643201	2219	99.9
M1_4	*Nocardia* sp. strain OAct 132	JX047071	2671	99.9
M1_7	*Arthrobacter* sp. strain 3S-5	KM434250	2670	99.9
M1_9	*Tomitella biformata* strain AHU 1821	NR_112905	2575	98.9
M2_1	*Arthrobacter* sp. (uncultured clone)	KJ650689	2671	100
M2_11	*Frigoribacterium* sp. strain FB3	AM933497	2657	100
M2_15	*Rhodococcus jialingiae* strain djl-6-2 16S	NR_115708	2675	99.9
M2_4	*Arthrobacter* sp. strainRKS6-4	GQ477171	2670	99.9
M2_9	*Streptomyces* sp. strain MM56	KU714908	2684	100
M3_10	*Streptomyces* sp. strain MM56	KU714908	2679	99.9
M3_8	*Arthrobacter* sp. strain 3S-5	KM434250	2647	99.7
M3_9	*Arthrobacter* sp. strain MNPB6	FM213396	2555	98.3
M4_18	*Rhodococcus maanshanensis* strain GMC121	AB741451	2465	97.6
M4_21	*Arthrobacter* sp. strain EM0174	HM165266	2559	98.5
M4_24	*Streptomyces* sp. strain MM56	KU714908	2679	99.9
M4_9	*Nocardiopsis umidischolae* strain NBRC 100349	NR_112746	2690	100
M5_2	*Nocardia* sp. strain OAct 132	JX047071	2644	99.6
M5_6	*Nocardia* sp. strain OAct 132	JX047071	2633	99.4
M5_8	*Streptomyces* sp. strain MM56	KU714908	2694	100
M5_9	*Streptomyces* sp. strain MM56	KU714908	2675	99.9
W2_1	*Microbacterium phyllosphaerae* IHBB 11136	KR085857	2686	100

^1^ GenBank accession numbers for 16S rDNA sequences of the isolates are KU643201.1, KU643207.1, MG758033.1, and MG758033.1. ^2^ GenBank accession numbers for genomic sequences of the organisms with the strongest 16S rDNA sequence matches to the isolates are provided.

**Table 2 antibiotics-10-01212-t002:** Bacterial and fungal strains used for determination of antimicrobial activities of isolated *Actinobacteria* from the Szczelina Chochołowska cave.

Bacterial or Fungal Strain	Source
*Staphylococcus aureus* MRSA 200	Medical University of Gdańsk
*Staphylococcus aureus* MRSA ATCC 6538	Medical University of Gdańsk
*Staphylococcus aureus* MRSA 108	Medical University of Gdańsk
*Staphylococcus aureus* MRSA 271	Medical University of Gdańsk
*Staphylococcus aureus* MRSA 203	Medical University of Gdańsk
*Staphylococcus aureus* MRSA 122	Medical University of Gdańsk
*Staphylococcus aureus* MRSA 116	Medical University of Gdańsk
*Staphylococcus aureus* MRSA 115	Medical University of Gdańsk
*Staphylococcus aureus* MRSA 342	Medical University of Gdańsk
*Staphylococcus aureus* MRSA 352	Medical University of Gdańsk
*Staphylococcus aureus* MRSA 44	Medical University of Gdańsk
*Staphylococcus aureus* MRSA 298	Medical University of Gdańsk
*Staphylococcus aureus* MRSA 199	Medical University of Gdańsk
*Staphylococcus aureus* MRSA 343	Medical University of Gdańsk
*Staphylococcus aureus* MRSA 297	Medical University of Gdańsk
*Staphylococcus aureus* MRSA 202	Medical University of Gdańsk
*Staphylococcus aureus* MRSA 124	Medical University of Gdańsk
*Staphylococcus aureus* MRSA 149	Medical University of Gdańsk
*Salmonella enterica* Virchow 41	National Salmonella Center, Gdańsk, Poland
*Salmonella enterica* Enteritidis 64	National Salmonella Center, Gdańsk, Poland
*Salmonella enterica* Kentucky 1368	National Salmonella Center, Gdańsk, Poland
*Salmonella enterica* Heidelberg 16	National Salmonella Center, Gdańsk, Poland
*Salmonella enterica* Cholerasuis 1439	National Salmonella Center, Gdańsk, Poland
*Salmonella enterica* Typhimurium 12	National Salmonella Center, Gdańsk, Poland
*Salmonella enterica* Typhimurium 13	National Salmonella Center, Gdańsk, Poland
*Salmonella enterica* Agona 1408	National Salmonella Center, Gdańsk, Poland
*Salmonella enterica* Thompson 39	National Salmonella Center, Gdańsk, Poland
*Salmonella enterica* Gallinarum 74	National Salmonella Center, Gdańsk, Poland
*Salmonella enterica* Hadar 1784	National Salmonella Center, Gdańsk, Poland
*Salmonella enterica* Cholerasuis 39	National Salmonella Center, Gdańsk, Poland
*Salmonella enterica* Infantis 155	National Salmonella Center, Gdańsk, Poland
*Salmonella enterica* Bovismorbificans 300	National Salmonella Center, Gdańsk, Poland
*Salmonella enterica* Seftenberg 87	National Salmonella Center, Gdańsk, Poland
*Salmonella enterica* Newport 50	National Salmonella Center, Gdańsk, Poland
*Salmonella enterica* Newport 51	National Salmonella Center, Gdańsk, Poland
*Salmonella enterica* Cholerasuis 37	National Salmonella Center, Gdańsk, Poland
*Salmonella enterica* Dubin 65	National Salmonella Center, Gdańsk, Poland
*Salmonella enterica* Saindpaul 435	National Salmonella Center, Gdańsk, Poland
*Salmonella enterica* Enteritidis 1392	National Salmonella Center, Gdańsk, Poland
*Escherichia coli* STEC 35	University of Gdańsk collection
*Escherichia coli* STEC 36	University of Gdańsk collection
*Escherichia coli* STEC 37	University of Gdańsk collection
*Escherichia coli* STEC 38	University of Gdańsk collection
*Escherichia coli* STEC 39	University of Gdańsk collection
*Pseudomonas aeruginosa* 02113	University of Gdańsk collection
*Pseudomonas aeruginosa* 02109	University of Gdańsk collection
*Pseudomons aeruginosa* 02108	University of Gdańsk collection
*Pseudomonas aeruginosa* RA743	University of Gdańsk collection
*Bacillus subtilis* 3610	University of Gdańsk collection
*Bacillus subtilis* wt168	University of Gdańsk collection
*Bacillus megaterium*	University of Gdańsk collection
*Bacillus cereus*	University of Gdańsk collection
*Candida parapsilosis* D2	Bruss Laboratories, Gdynia, Poland
*Candida glabrata* D3	Bruss Laboratories, Gdynia, Poland
*Candida tropicalis* D4	Bruss Laboratories, Gdynia, Poland
*Candida dubliniensis* D5	Bruss Laboratories, Gdynia, Poland
*Candida albicans* D6	Bruss Laboratories, Gdynia, Poland
*Candida albicans* D7	Medical University of Gdańsk
*Candida albicans* D8	Medical University of Gdańsk
*Candida albicans* D9	University Clinical Centre in Gdańsk
*Candida albicans* E1	University Clinical Centre in Gdańsk
*Candida guilliermondii* E2	University Clinical Centre in Gdańsk
*Candida guilliermondii* E3	University Clinical Centre in Gdańsk
*Candida albicans* E4	University Clinical Centre in Gdańsk
*Candida albicans* E5	University Clinical Centre in Gdańsk
*Candida glabrata* E6	University Clinical Centre in Gdańsk
*Candida glabrata* E7	University Clinical Centre in Gdańsk
*Candida* sp. E8	University Clinical Centre in Gdańsk
*Candida* sp. E9	University Clinical Centre in Gdańsk

**Table 3 antibiotics-10-01212-t003:** Effects of *Streptomyces* M4_24 and M5_8 isolates on growth inhibition of bacterial strains. The width of the inhibition zone was measured in the streak-test after 48 h of incubation.

Bacteruial Strain	Growth Inhibition Zone (mm) ^1^	
	M4_24	M5_8
*S. aureus* MRSA 200	5.0 ± 1.0	5.5 ± 1.5
*S. aureus* MRSA ATCC 6538	7.5 ± 0.3	6.5 ± 0.5
*S. aureus* MRSA 108	5.0 ± 2.0	6.5 ± 0.5
*S. aureus* MRSA 271	7.5 ± 1.5	7.5 ± 0.5
*S. aureus* MRSA 203	6.5 ± 1.5	7.0 ± 1.0
*S. aureus* MRSA 122	4.5 ± 0.5	4.75 ± 1.25
*S. aureus* MRSA 116	5.0 ± 1.0	5.5 ± 0.5
*S. aureus* MRSA 115	6.75 ± 0.75	5.75 ± 0.75
*S. aureus* MRSA 342	0.0	0.0
*S. aureus* MRSA 352	6.0 ± 1.0	5.25 ± 1.75
*S. aureus* MRSA 44	12.0 ± 3.0	0.0
*S. aureus* MRSA 298	4.0 ± 0.0	0.0
*S. aureus* MRSA 199	8.0 ± 2.0	0.0
*S. aureus* MRSA 343	8.0 ± 1.0	0.0
*S. aureus* MRSA 297	7.25 ± 1.25	6.75 ± 0.25
*S. aureus* MRSA 202	8.25 ± 1.25	3.5 ± 0.5
*S. aureus* MRSA 124	3.0 ± 0	3.0 ± 0.0
*S. aureus* MRSA 149	6.0 ± 0.0	6.0 ± 0.0
*S. enterica* Virchow 41	0.0	0.0
*S. enterica* Enteritidis 64	6.0 ± 1.0	0.0
*S. enterica* Kentucky 1368	5.5 ± 0.5	8.0 ± 1.0
*S*. *enterica* Heidelberg 16	6.0 ± 1.0	5.0 ± 1.0
*S. enterica* Cholerasuis 1439	11.5 ± 0.5	0.0
*S. enterica* Typhimurium 12	7.5 ± 1.5	0.0
*S. enterica* Typhimurium 13	6.5 ± 0.5	0.0
*S. enterica* Agona 1408	0.0	0.0
*S. enterica* Thompson 39	0.0	0.0
*S. enterica* Gallinarum 74	5.5 ± 0.5	0.0
*S. enterica* Hadar 1784	0.0	0.0
*S. enterica* Cholerasuis 39	6.5 ± 1.5	5.0 ± 2.0
*S. enterica* Infantis 155	6.5 ± 0.5	8.0 ± 10
*S. enterica* Bovismorbificans 300	6.25 ± 0.25	0.0
*S. enterica* Seftenberg 87	5.0 ± 1.0	4.5 ± 0.5
*S. enterica* Newport 50	5.5 ± 0.5	5.0 ± 1.0
*S. enterica* Newport 51	5.5 ± 0.5	5.0 ± 1.0
*S. enterica* Cholerasuis 37	4.5 ± 0.5	6.5 ± 0.5
*S. enterica* Dubin 65	6.5 ± 0.5	4.5 ± 0.5
*S. enterica* Saindpaul 435	3.0 ± 1.0	0.0
*S. enterica* Enteritidis 1392	9.75 ± 0.25	4 ± 0
*Enterococcus sp.*	10.5 ± 0.5	8.0 ± 1.0
*E. coli* 35	8.5 ± 1.5	6.5 ± 0.5
*E. coli* 36	10.5 ± 1.5	7.0 ± 1.0
*E. coli* 37	10.0 ± 1.0	7.0 ± 0.5
*E. coli* 38	11.0 ± 0.5	6.5 ± 0.5
*E. coli* 39	8.5 ± 0.5	8.5 ± 0.5
*B. subtilis* 3610	7.0 ± 1.0	0.0
*B. subtilis* wt168	8.0 ± 1.0	0.0
*B. megaterium*	8.0 ± 1.0	0.0
*B. cereus*	7.0 ± 2.0	0.0
*P. aeruginosa* 02113	8.0 ± 2.0	6.0 ± 1.0
*P. aeruginosa* 02109	6.5 ± 0.5	5.0 ± 0.5
*P. aeruginosa* 02108	10.0 ± 1.0	5.0 ± 0.0
*P. aeruginosa* RA743	8.0 ± 1.0	7.0 ± 2.0

^1^ Mean values from 3 independent experiments ± SD are demonstrated.

**Table 4 antibiotics-10-01212-t004:** Effects of *Streptomyces* M4_24 and M5_8 isolates on growth inhibition of fungal strains. The width of the inhibition zone was measured in the streak-test after 48 h of incubation.

Fungal Strain	Growth Inhibition Zone (mm) ^1^	
	M4_24	M5_8
*Candida parapsilosis* D2	0.0	0.0
*Candida glabrata* D3	10.3 ± 2.1	18.7 ± 3.5
*Candida tropicalis* D4	0.0	0.0
*Candida dubliniensis* D5	5.7 ± 1.5	0.0
*Candida albicans* D6	3.3 ± 1.2	0.0
*Candida albicans* D7	3.7 ± 0.6	3.0 ± 1.0
*Candida albicans* D8	0.0	0.0
*Candida albicans* D9	4.3 ± 1.5	2.3
*Candida albicans* E1	3.3 ± 2.3	0.0
*Candida guilliermondii* E2	6.0 ± 2.6	0.0
*Candida guilliermondii* E3	6.3 ± 0.6	0.0
*Candida albicans* E4	3.3 ± 0.6	0.0
*Candida albicans* E5	0.0	0.0
*Candida glabrata* E6	12.0 ± 4.4	5.7 ± 2.5
*Candida glabrata* E7	6.0 ± 2.0	2.0 ± 3.5
*Candida* sp. E8	3.7 ± 0.6	3.7 ± 1.2
*Candida* sp. E9	3.3 ± 1.2	3.3 ± 1.5

^1^ Mean values from 3 independent experiments ± SD are demonstrated.

## Data Availability

DNA sequences determined in this study were deposited in GenBank with accession numbers KU643201.1, KU643207.1, MG758033.1, and MG758033.1.

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
