# Peer review of "Antibacterial, Antifungal and Anticancer Activities of Compounds Produced by Newly Isolated Streptomyces Strains from the Szczelina Chochołowska Cave (Tatra Mountains, Poland)"

_antibiotics, 2021, doi:10.3390/antibiotics10101212_

Round 1

Reviewer 1 Report

1 Anthrabenzoxocinone compounds are a series of important natural antibiotics. The author may give more discussion on the activities and natural sources of this class of antibiotics.

2 This study found Bacteriophage virions in samples derived from the Nocardia sp. strain JSZCL7, but this strain did not display antibacterial, antifungal or anticancer activities in the present study. How could the authors conclude that “Nevertheless, the presence of bacteriophages in samples of isolated bacteria may suggest that these viruses might influence production of bioactive compounds by these strains.”? (page 5, line 133)

Author Response

REVIEWER’S COMMENT:

1 Anthrabenzoxocinone compounds are a series of important natural antibiotics. The author may give more discussion on the activities and natural sources of this class of antibiotics.

RESPONSE:

We thank the reviewer for this comment. This aspect has been discussed in more details, and the following text has been added to Discussion (lines: 277-296):

“It seems that anthrabenzoxocinone compounds might be of special interested among substances isolated from Streptomyces strains. Early studies on Streptomyces violaceusniger, isolated in Japan, led to discovery of a potent anthrabenzoxocinone derivative, named BE-24566B, which might be used against MRSA [39]. Two such compounds were then isolated from Streptomyces sp. (MA6657), and their characteristics indicated that they have significant antimicrobial activities against Gram-negative bacteria, with MIC values ranging from 0.5 to 2.0 g/ml [12]. Genetic analysis of Streptomyces sp. FJS31-2 indicated that its genome contains a gene cluster which is responsible for production of anthra-benzoxocinones with potential antibacterial activities, previously known BE-24566B and newly identified zunyimycin A [40]. The same Streptomyces strain, when cultured under various laboratory conditions, produced other compounds from this group, named zunyimycins B and C, inhibiting growth of both MRSA and enterococci [41]. These results demonstrated a high potential of Actinobacteria to produce different antimicrobial com-pounds which may be modified under different environmental conditions. In addition, antibiotic activities of natural anthrabenzoxocinones can be enhanced by biochemical modifications [42]. Furthermore, recent studies highlighted a high biodiversity of natural products, including anthrabenzoxocinone compounds, produced by bacteria isolated from natural habitats [28], and a high genetic potential of Streptomycs spp. in production of variety of bioactive compounds, also including anthrabenzoxocinones [43]. Our results, presented in this report, corroborate these conclusions.”

REVIEWER’S COMMENT:

2 This study found Bacteriophage virions in samples derived from the Nocardia sp. strain JSZCL7, but this strain did not display antibacterial, antifungal or anticancer activities in the present study. How could the authors conclude that “Nevertheless, the presence of bacteriophages in samples of isolated bacteria may suggest that these viruses might influence production of bioactive compounds by these strains.”? (page 5, line 133)

RESPONSE: We agree that this statement was too strong. We have toned it down, by modifying the sentence as follows (lines: 133-136):

“Nevertheless, the presence of bacteriophages in samples of isolated bacteria may suggest that this group of viruses might be taken into consideration in further studies on bioactive compound-producing Actinobacteria, as putative modulators of syntheses of such substances.”

Reviewer 2 Report

The topic of this manuscript is interesting although the chemical structure and  mechanism is unclear. The reviewer feels it can be accepted after some minor amendments.

(1) Do the chemical listed in Fig 9 display anti-microbial and/or anti-cancer effects?

(2) How much Streptomyces extract was used in the in vitro study? % is not a good unit.

(3)  Do such chemical display any selectivity, ie  less toxic to non-transformed cells?

Author Response

REVIEWER’S COMMENT:

The topic of this manuscript is interesting although the chemical structure and  mechanism is unclear. The reviewer feels it can be accepted after some minor amendments.

  • Do the chemical listed in Fig 9 display anti-microbial and/or anti-cancer effects?

RESPONSE:

At the current stage of our knowledge, we can only indicate that the isolated Actinobacteria (Streptomyces strains) and extracts containing these compounds reveal antibacterial, antifungal and anti-cancer activities. Further studies are required to test specific activities of separate compounds. We were not able to obtain sufficient amounts of these compounds to perform such experiments, thus, either conditions of their production by isolated strains should be optimized, or chemical synthesis of these compounds can be considered in the future.

REVIEWER’S COMMENT:

  • How much Streptomyces extract was used in the in vitro study? % is not a good unit.

RESPONSE:

We appreciate this comment, however, since the extracts are mixtures of many compounds, it is difficult to indicate any other amount units apart from percentage of the extract added to a tested sample. In fact, a precise description of preparation of extracts (as provided in Sections 4.7 and 4.10) allows to estimate what does the percentage actually represent.

REVIEWER’S COMMENT:

  • Do such chemical display any selectivity, ie  less toxic to non-transformed cells

RESPONSE:

We thank reviewer for this important comment. In fact, we have performed experiments with non-transformed cells, and their results are described in the revised manuscript as follows: (lines: 180-182):

“In control experiments, we did not observe any significant reduction of viability of non-transformed cells, the HDFa cell line, except conditions of 100% extract”.

Reviewer 3 Report

Jaroszewicz et al have isolated several strains of Streptomyces and tested the extracts for antifungal, antibacterial and cytotoxic properties together with the identification of some chemical components in the extracts using LC MS. The manuscript was written well including chemical analysis of the identified compounds. 

However, the previous isolation of the identified chemical compounds as natural products should be more discussed to confirm its production from Streptomyces.

More comments:

Figure 7 line 217 and figure 8 line 219 should be labelled correctly using the compounds name or the molecular weight

Author Response

REVIEWER’S COMMENT:

Jaroszewicz et al have isolated several strains of Streptomyces and tested the extracts for antifungal, antibacterial and cytotoxic properties together with the identification of some chemical components in the extracts using LC MS. The manuscript was written well including chemical analysis of the identified compounds. 

However, the previous isolation of the identified chemical compounds as natural products should be more discussed to confirm its production from Streptomyces.

RESPONSE:

A more detailed discussion on previously isolated natural anthrabenzoxocinones, and their modifications, is presented in the revised manuscript. The following text has been added to Discussion (lines 277-296):

“It seems that anthrabenzoxocinone compounds might be of special interested among substances isolated from Streptomyces strains. Early studies on Streptomyces violaceusniger, isolated in Japan, led to discovery of a potent anthrabenzoxocinone derivative, named BE-24566B, which might be used against MRSA [39]. Two such compounds were then isolated from Streptomyces sp. (MA6657), and their characteristics indicated that they have significant antimicrobial activities against Gram-negative bacteria, with MIC values ranging from 0.5 to 2.0 g/ml [12]. Genetic analysis of Streptomyces sp. FJS31-2 indicated that its genome contains a gene cluster which is responsible for production of anthra-benzoxocinones with potential antibacterial activities, previously known BE-24566B and newly identified zunyimycin A [40]. The same Streptomyces strain, when cultured under various laboratory conditions, produced other compounds from this group, named zunyimycins B and C, inhibiting growth of both MRSA and enterococci [41]. These results demonstrated a high potential of Actinobacteria to produce different antimicrobial com-pounds which may be modified under different environmental conditions. In addition, antibiotic activities of natural anthrabenzoxocinones can be enhanced by biochemical modifications [42]. Furthermore, recent studies highlighted a high biodiversity of natural products, including anthrabenzoxocinone compounds, produced by bacteria isolated from natural habitats [28], and a high genetic potential of Streptomycs spp. in production of variety of bioactive compounds, also including anthrabenzoxocinones [43]. Our results, presented in this report, corroborate these conclusions.”

REVIEWER’S COMMENT:

Figure 7 line 217 and figure 8 line 219 should be labelled correctly using the compounds name or the molecular weight

RESPONSE:

The Figures were modified according to the reviewer’s recommendations.

Reviewer 4 Report

This reviewer has some minor comments for authors to improve their manuscript before acceptance in Antibiotics.

1) In Fig. 1, it might be better to write "Poland" into the map (top one). Scale bar is not clear to be identified in B and D. Same issues are also in Fig. 2, and "Scale bar = XX nm" should be added.

2) It is interesting to spot bacteriophages in the samples. Further identification using the genome-seq of the bacterial pool might be needed. By acquiring this information, the accuracy of table 2 can also be improved. 

3) A set of cultured plates for inhibition assay (Tbls 3&4) should be provided as supplemental results. Those are the essential records for this manuscript. 

4) A scanned picture of the 96-wall plate (?) for the MTT test as summarized in Fig. 4 and Fig. 5, at least for Fig. 5.

5) Image quality for HPLC and MS should be improved in the revised version. It seems fine to combine 7&8. Label for X/Y-axis is required.

Author Response

REVIEWER’S COMMENT:

1) In Fig. 1, it might be better to write "Poland" into the map (top one). Scale bar is not clear to be identified in B and D. Same issues are also in Fig. 2, and "Scale bar = XX nm" should be added.

RESPONSE:

Figures 1 and 2 were modified as recommended by the reviewer.

REVIEWER’S COMMENT:

2) It is interesting to spot bacteriophages in the samples. Further identification using the genome-seq of the bacterial pool might be needed. By acquiring this information, the accuracy of table 2 can also be improved. 

RESPONSE:

We have spotted bacteriophages in the samples, however, as indicated in the manuscript (lines 127-130), we were unable to identify a phage-sensitive strain (obviously, the host lysogenic strain was also resistant to this bacteriophage). We agree that genome sequence data would be very interesting and informative, however, since this is a significant amount of work, we assessed that such analyses would be rather a good material for a separate article in the future.

REVIEWER’S COMMENT:

3) A set of cultured plates for inhibition assay (Tbls 3&4) should be provided as supplemental results. Those are the essential records for this manuscript. 

RESPONSE:

As suggested by the reviewer, cultured plates with the inhibition assay are demonstrated as supplementary results (Supplementary Figure S1).

REVIEWER’S COMMENT:

4) A scanned picture of the 96-wall plate (?) for the MTT test as summarized in Fig. 4 and Fig. 5, at least for Fig. 5.

RESPONSE:

Since the MTT test is a quantitative assay, based on quantitative measurement of absorbance at the final step, we decided to present quantitative values, normalized to controls (as shown in Figures 4 and 5), rather than pictures of plates with more or less intensive colors which indicate only roughly the results of experiments.

REVIEWER’S COMMENT:

5) Image quality for HPLC and MS should be improved in the revised version. It seems fine to combine 7&8. Label for X/Y-axis is required.

RESPONSE:

As requested by the reviewer, quality of images presented UHPLC and LC-MS was significantly improved. Moreover, axes have been labelled and captions of figures have been modified to be more precise and informative.